

**Hydrological response to climate change and human activities in the Three-River Source Region**

Ting Su[1], Chiyuan Miao*[1], Qingyun Duan[2], Jiaojiao Gou[1], Xiaoying Guo[1], Xi Zhao[1]

[1] State Key Laboratory of Earth Surface Processes and Resource Ecology, Faculty of

Geographical Science, Beijing Normal University, Beijing 100875, China

[2] College of Hydrology and Water Resources, Hohai University, Nanjing 210024, China

*Correspondence to*: Chiyuan Miao (miaocy@vip.sina.com)

**Abstract.** The Three-River Source Region (TRSR), which is known as "China's Water Tower" and affects the water resources security of 700 million people living

downstream, has experienced significant hydrological changes in the past few decades. In this work, we used an extended variable infiltration capacity (VIC) land surface hydrologic model (VIC-Glacier) coupled with the degree-day factor algorithm to simulate the runoff change in the TRSR during 1984–2018. VIC-Glacier performed well in the TRSR, with Nash-Sutcliffe efficiency (*NSE*) above 0.68, but it was sensitive

to the quality of the limited ground-based precipitation. This was especially marked in the source region of the Yangtze River: when we used Precipitation Estimation from Remotely Sensed Information Using Artificial Neural Networks – Climate Data Record (PERSIANN-CDR), which has better spatial details, instead of ground-based precipitation, the *NSE* of Tuotuohe Station increased from 0.31 to 0.86. Using the

well-established VIC-Glacier model, we studied the contribution of each runoff component (rainfall, snowmelt, and glacier runoff) to the total runoff and the causes of changes in runoff. The results indicate that rainfall runoff contributed over 80% of the total runoff, while snowmelt runoff and glacier runoff both contributed less than 10% in 1984–2018. Climate change was the main reason for the increase in runoff in the

TRSR after 2004, accounting for 75%–89%, except in the catchment monitored by Xialaxiu Station. Among climate change factors, precipitation had the greatest impact





on runoff. Finally, through a series of hypothetical climate change scenario experiments, we found that a future simultaneous increase in precipitation and temperature would increase the total runoff, rainfall runoff, and glacier runoff. The snowmelt runoff might

remain unchanged, because the increased precipitation, even with seasonal fluctuations, was basically completely compensated for by the decreased solid-to-liquid precipitation ratio. These findings improve our understanding of hydrological processes and provide insights for policy makers on how to optimally allocate water resources and manage the TRSR in response to global climate change.

**1 Introduction**

Known as the "Asian Water Tower" (Immerzeel et al., 2010) and the "Third Pole" (Qiu, 2008), the Qinghai-Tibet Plateau (QTP) is the source of many large Asian rivers (e.g., the Yellow, Yangtze, Salween, Mekong, Ganges, Indus, and Yarlung Zangbo Rivers) (Cuo et al., 2019), and it supports diverse ecosystems and affects the survival and

development of more than one billion people living downstream (Qiu, 2008). The QTP is also the largest repository of glaciers, snow, and frozen soil outside of the Arctic and Antarctic, and it is extremely sensitive to climate change (Yao, 2019; Gao et al., 2019; Wang et al., 2021a). In recent decades, the temperature of the QTP has risen significantly, with the warming rate double the global average level in the same period,

and the precipitation has also increased overall (Yao, 2019; Xu et al., 2008). As its climate has become warmer and wetter, the QTP's hydrological cycling has been correspondingly enhanced in terms of glacier retreat (Gao et al., 2019; Zhu et al., 2022), snowpack reduction (Huang et al., 2017), permafrost degradation (Cuo et al., 2015; Liu et al., 2020), and lake expansion (Zhang et al., 2017a; Liu et al., 2020), and these

changes are expected to intensify under future continued warming (Immerzeel et al., 2013). However, how runoff in the glaciated regions of the QTP will change is controversial: it could initially increase as a result of accelerated glacier melting but eventually decrease due to the loss of glacier area (Immerzeel et al., 2013; Su et al., 2016; Zhao et al., 2019; Barnett et al., 2005). All these changes increase the uncertainty

of regional water resources simulation and prediction, thus posing great challenges to





the scientific management and rational distribution of water resources.

Located in the hinterland of the QTP (Liu et al., 2017a), the Three-River Source Region (TRSR) is the headwaters of the Yellow, Yangtze, and Lancang Rivers, and is known as

"China's Water Tower" (Ji and Yuan, 2018a). Approximately 49%, 20%, and 15% of the total water volume of the Yellow, Yangtze, and Lancang Rivers, respectively, is provided by the TRSR (Cao and Pan, 2014). Thus, the TRSR plays an extremely important role in water resources security and ecological and environmental protection in China and even all of Southeast Asia (Zhang et al., 2019). As on the QTP, during the

past few decades the climate has changed in the TRSR, with precipitation and temperature, respectively, increasing at rates of 6.653–10.31 mm·10a$^{-1}$ and 0.33 ℃·10a$^{-1}$ (Cai et al., 2022; Meng et al., 2020). These changes, coupled with intensive human activities, especially the ecological restoration and protection projects initiated by the Chinese government in this century (Zhang et al., 2017b; Liu et al.,

2017a), have led to changes in runoff. These are, mainly, significantly increased runoff in the source regions of the Yangtze and Lancang Rivers and a weak downward trend in the source region of the Yellow River in the past few decades (Meng et al., 2020; Zhou and Huang, 2012). Most researchers have attributed the increased runoff in the source regions of the Yangtze and Lancang Rivers to climate change, with its

contribution exceeding 90% (Jiang et al., 2016; Ahmed et al., 2021), but why runoff has declined at the source of the Yellow River is a subject of dispute. Zheng et al. (2009) and Feng et al. (2017) found that land use change has played a more important role in reducing runoff in the source region of the Yellow River, while others have reported that 86% of the runoff reduction could be attributed to climate change, including natural

and anthropogenic climate change (Ji and Yuan, 2018b). Most of these studies have focused on the overall impact of climate change and have not examined the possible influence of specific and single climatic variables (e.g., precipitation and temperature) on runoff. And because they only selected one or a few hydrological stations in each source region, the spatial heterogeneity of runoff change in response to multiple factors

was not well considered. Therefore, we need to study the causes of changes in runoff



across the entire TRSR in more detail.

Meltwater from cryosphere elements such as glaciers and snow is an important source of runoff in the TRSR (Han et al., 2019; Meng et al., 2020). Accurate observation or
simulation of snow/glacier meltwater is crucial in understanding the hydrological cycle and managing water resources. However, few distributed hydrologic models have been specifically designed for alpine regions and to consider the complexity of runoff generation (Zhao et al., 2012; Yang et al., 2012), and the existing hydrologic models mostly ignore glacial melting (Zhao et al., 2012; Shangguan et al., 2015; Zhang et al.,
2013). In addition, the inclusion of meteorological forcings, especially precipitation, is essential for reliable hydrological simulation (Liu et al., 2017b; Sun and Su, 2020). Because of the high altitude and complicated terrain, and the high cost of establishing and maintaining meteorological stations in the harsh environment, such stations are sparse, and therefore the limited rain-gauge interpolation data they provide may not
accurately reflect the distribution of precipitation, which displays great temporal and spatial variability in the TRSR (Sun and Su, 2020; Ji and Yuan, 2018a). Some of these stations also lack long-term flow observation records (Wang et al., 2021a), making the hydrological modeling of these regions more difficult. Although published studies have provided some insights into the separate components of runoff in the TRSR, no
consistent conclusion has been reached. For example, Zhang et al. (2013) estimated that in the source region of the Yangtze River, snowmelt runoff and glacier runoff, respectively, accounted for 22.2% and 6.5% of the total runoff during 1961–2009, while Han et al. (2019) found that these two together accounted for only about 12% in 2003–2014. In the source region of the Yellow River, the total proportion of glacier and
snowmelt runoff also varied from 17% to 23.2% in different periods (Wang et al., 2021b; Zhang et al., 2013; Zhang et al., 2022). These discrepancies lead us to conclude that there is still a lack of systematic research on the contributions of specific runoff components over the TRSR. To better understand the impact of future climate change on runoff, hydrologic models driven by hypothetical climate change scenarios or
climate model projections have long been commonly used to evaluate the hydrological





consequences of climate change (Su et al., 2016). To date, however, insufficient attention has been given to developing a comprehensive understanding of the TRSR.

In this study, we used the variable infiltration capacity (VIC) land surface hydrologic model linked with the degree-day factor algorithm to simulate the runoff change in the TRSR, aiming to address the following objectives: (1) Quantify the runoff components (rainfall runoff, snowmelt runoff, and glacier runoff) in the TRSR. (2) Separate the impacts of climate change and human activities on runoff change. (3) Analyze the responses of total runoff and runoff components under the hypothetical climate change scenarios. We expect the results will help to guide current and future regulation and management of water resources in the TRSR.

## 2 Study area, data sources, and methods

### 2.1 Study area

The TRSR (30 °N–36 °N, 90 °E–104 °E) is located in the hinterland of the QTP, with an altitude ranging from 2,677 m to 6,575 m (Figure 1). It is the headwaters of the Yangtze, Yellow, and Lancang Rivers, covering an area of approximately $36.1 \times 10^4 \ \mathrm{km}^2$ (Ji et al., 2020). The TRSR's climate is a typical plateau continental climate, characterized by low temperatures, strong radiation, and no obvious distinction between four seasons (Tong et al., 2010). During 1981–2010, the annual mean precipitation was 593 mm and the annual mean temperature was 1.9 °C on the TRSR (Luo et al., 2017), which both generally decreased from southeast to northwest (Deng and Zhang, 2018). Approximately 75% of total annual precipitation occurs from June to September, and the temperature is always highest during this period (Figure S1). The average annual runoff within the TRSR is about $47.5 \times 10^9 \ \mathrm{m}^3$. Glaciers are widely distributed in the TRSR, with about 1,700 glaciers covering an area of about 2,300 $\mathrm{km}^2$ in all.

**Figure 1**

### 2.2 Data sources

Three kinds of data were included in this study, namely, meteorological forcing data,





land surface characteristic data, and runoff data. The meteorological forcing data—
including daily precipitation, wind speed, and maximum, minimum, and mean temperatures from 1983 to 2018—were obtained from the China Meteorological Administration (CMA) (http://data.cma.cn). Most weather stations are located in the south and southeast of the TRSR, with very few in the central and western regions (Figure 1). For the Yangtze River source region, the interpolated precipitation based on
CMA rain gauges (hereinafter called CMA precipitation) cannot capture the spatial details well (Liu et al., 2017b; Xue et al., 2013). Therefore, in this sub-region, we also used Precipitation Estimation from Remotely Sensed Information Using Artificial Neural Networks – Climate Data Record (PERSIANN-CDR) (ftp://data.ncdc.noaa.gov/cdr/persiann/files/), which has the advantages of high spatial
$(0.25° \times 0.25°)$ and temporal (daily) resolution, long time span (more than 35 years since 1983, and the data range is constantly updated), and more complete coverage (Ashouri et al., 2015). Previous studies have also confirmed that PERSIANN-CDR is a high-quality precipitation data set applicable to the source region of the Yangtze River (Liu et al., 2017b; Wang et al., 2021b).


The soil texture data came from the global 5-arcmin data set of the Food and Agriculture Organization of the United Nations (FAO), and the vegetation types were provided by the global 1-km land cover classification database produced by the University of Maryland (http://glcfapp.glcf.umd.edu:8080/esdi/index.jsp). The Shuttle Radar
Topography Mission (SRTM) digital elevation data set with a resolution of 90 m was obtained from the Geospatial Data Cloud (http://www.gscloud.cn/search). Glacier area data were from Ye et al. (2017), and we assumed that the glacier area around the source of the Yellow River was 0 km$^2$ due to its proportion of the whole area and contribution to total runoff being minor (Wang et al., 2021b; Zhang et al., 2013).


Monthly runoff observation data were from the corresponding hydrological stations. The details of the time ranges of the available data are shown in Table 1. It should be noted that there was no observed runoff from November to April at Tuotuohe Station





and that, in this study, we treated the runoff in these months as 0, because the runoff

was negligible (Ahmed et al., 2020; Luo et al., 2019).

**Table 1**

**2.3 Methods**

**2.3.1 VIC-Glacier model implementation**

The VIC model is a large-scale, distributed land surface hydrologic model (Liang et al.,

1994), and it can be used to simulate the balance of surface water and energy within

each grid cell at daily or sub-daily time steps, considering the processes of snow

accumulation, snow melting, and soil freezing and thawing (Liang et al., 1996;

Cherkauer and Lettenmaier, 2003). Surface runoff and base flow in each grid cell are

eventually routed to a specific watershed outlet by the Lohmann routing module

(Lohmann et al., 1996). However, the VIC model does not take glacier hydrological

processes into account (Zhang et al., 2013). Therefore, in this study, we coupled the

degree-day factor algorithm (Hock, 2003) to the VIC model to simulate the contribution

of glacier runoff to total runoff; the extended model is called VIC-Glacier. We ran the

VIC-Glacier model with a 24-hour time step at 0.25 ° spatial resolution for a period of

36 years (1983–2018) and set a one-year warm-up period to get the ideal initial state.

The total runoff and runoff components of each grid cell can be calculated as:

$$R_i = f_i \times G_i + (1 - f_i) \times R_{vic,i} \tag{1}$$

$$R_{rainfall,i} = R_{vic,i} - R_{snowmelt,i} \tag{2}$$

where $R_i$ is the total runoff (mm) of grid cell $i$, $R_{vic,i}$ is the runoff (mm) of grid cell $i$

calculated by the original VIC model, $G_i$ is the glacier runoff (mm), $f_i$ is the glacier

area fraction (%) of grid cell $i$, and $R_{rainfall,i}$ and $R_{snowmelt,i}$ are the rainfall runoff

(mm) and snowmelt runoff (mm) of grid cell $i$, respectively. The $G_i$ can be calculated

as:

$$G_i = P_i + M_i \tag{3}$$

$$M_i = \begin{cases} DDF \times T_i, & T_i > 0 \\ 0, & T_i \le 0 \end{cases} \tag{4}$$

where $P_i$ is the rainfall (mm) of grid cell $i$, $M_i$ is the glacier meltwater (mm) in grid



cell $i$, DDF is the degree-day factor for glaciers (mm $°C^{-1}$ $day^{-1}$) directly taken from Zhang et al. (2006), $T_i$ is the average daily temperature ( $°C$) of the glacier surface, and we adjust the temperature of the glacier area in each grid cell using the rate of
temperature decrease (0.65 $°C$/100 m).

Glacier volume affects the amount of melting ice. In this study, we used glacier volume to determine the maximum annual amount of ice melting (Liu et al., 2003), and the glacier volume was derived from the volume-area scaling relation (Bahr et al., 1997;
Radic et al., 2008):

$$V = 0.04 \times S^{1.43} \tag{5}$$

where $V$ is the glacier volume ($km^3$) and $S$ is the glacier area ($km^2$). The initial glacier volume of each grid cell was determined using the glacier area from the glacier distribution data set. After that, the volume was updated every year, and the updated
glacier area was determined from the updated glacier volume by using the inversion equation of formula (5). This process was repeated throughout the VIC-Glacier simulation until the glaciers were completely melted.

### 2.3.2 VIC-Glacier model optimization and calibration

Parameter optimization enables model simulations to be consistent with the
corresponding observations (Gupta et al., 1999). In this study, we considered 13 tunable runoff-related parameters (Table S1) and used an automatic calibration framework that combines sensitivity analysis and an adaptive surrogate modeling‐based optimization algorithm (Gou et al., 2020) to calibrate the parameters during the calibration period (1984–1993). The Nash-Sutcliffe efficiency coefficient ($NSE$) was used as an objective
function to describe the degree of matching between the simulated and observed values, and the model was considered to have performed well when $NSE$ was greater than 0.65. Next, Pearson's correlation coefficient ($R$) and relative error ($RE$) were calculated to evaluate the simulation performance:

$$NSE = 1 - \frac{\sum_{i=1}^{n}\left(R_{obs,i} - R_{sim,i}\right)^2}{\sum_{i=1}^{n}\left(R_{obs,i} - \bar{R}_{obs}\right)^2} \tag{6}$$




$$R = \frac{\sum_{i=1}^{n}(R_{obs,i}-\bar{R}_{obs})(R_{sim,i}-\bar{R}_{sim})}{\sqrt{\sum_{i=1}^{n}(R_{obs,i}-\bar{R}_{obs})^2}\sqrt{\sum_{i=1}^{n}(R_{sim,i}-\bar{R}_{sim})^2}} \qquad (7)$$

$$RE = \frac{\bar{R}_{sim}-\bar{R}_{obs}}{\bar{R}_{obs}} \times 100\% \qquad (8)$$

where $n$ is the number of monthly runoff series, $R_{obs,i}$ and $R_{sim,i}$ represent the observed and simulated runoff (mm) of the $i$th month, and $\bar{R}_{obs}$ and $\bar{R}_{sim}$ represent the average of observed and simulated values (mm), respectively. The closer the values of $NSE$ and $R$ are to 1, and the closer the value of $RE$ is to 0, the better the simulation results are.

The optimized parameters were then applied to the validation period (1994–2003) to verify whether the model was suitable for the study area by using the above-mentioned three indicators.

### 2.3.3 Attribution analysis

According to the degree of influence of human activities on the watershed environment, the whole runoff time series of the calibration and validation periods was regarded as the reference period (1984–2003), with less human activity, while the period after 2004 was taken as the changed period (2004–), with a greater effect of human activities on runoff. The observed change in runoff between these two periods reflects the joint influence of climate change and human activities. It can be specifically expressed as follows:

$$\Delta R_{total} = Rc_{obs} - Rr_{obs} = \Delta R_{human} + \Delta R_{climate} \qquad (9)$$

where $\Delta R_{total}$ (mm) indicates the observed change in mean annual runoff between these two periods; $Rc_{obs}$ (mm) and $Rr_{obs}$ (mm) are the average annual observed runoff in the changed period and the reference period, respectively; and $\Delta R_{climate}$ (mm) and $\Delta R_{human}$ (mm) are the changes in runoff caused by climate change and human activities, respectively.

The difference between the observed and natural runoff in the changed period reflects


the impact of human activities. The change in natural runoff between these two periods reflects the response to climate change, which can be further divided into precipitation-induced change, temperature-induced change and change induced by the interactions of climatic variables:

$$\Delta R_{human} = Rc_{obs} - (Rc_{sim} - \varepsilon) \tag{10}$$

$$\Delta R_{climate} = (Rc_{sim} - \varepsilon) - Rr_{obs} = \Delta R_P + \Delta R_T + \Delta R_{climate\_interactive} \tag{11}$$

$$\varepsilon = Rr_{sim} - Rr_{obs} \tag{12}$$

where $Rc_{sim}$ (mm) and $Rr_{sim}$ (mm) are the simulated average annual runoff in the changed period and the reference period, respectively, and $\varepsilon$ is the residual error, which may be related to factors not considered in this study, such as model simulation error. We assume that the residual errors for these two periods are the same. $\Delta R_P$ (mm) and $\Delta R_T$ (mm) represent the changes in runoff caused by precipitation change and temperature change, respectively. Due to the complex interactions of climatic factors in hydrological processes, as well as the fact that the influence of wind speed on runoff change was not discriminated, in this study we recorded the runoff change caused by climatic interactions and wind as $\Delta R_{climate\_interactive}$ (mm). $\Delta R_P$ and $\Delta R_T$ can be calculated as follows:

$$\Delta R_P = (R_{P_{sim}} - \varepsilon) - Rr_{obs} \tag{13}$$

$$\Delta R_T = (R_{T_{sim}} - \varepsilon) - Rr_{obs} \tag{14}$$

where $R_{P_{sim}}$ (mm) and $R_{T_{sim}}$ (mm) respectively represent the simulated runoff, which changes only precipitation or temperature to match the level in the changed period. Therefore, the percentage contribution of each factor to runoff change can be expressed as follows:

$$\omega_P = \frac{|\Delta R_P|}{|\Delta R_P| + |\Delta R_T| + |\Delta R_{climate\_interactive}| + |\Delta R_{human}|} \times 100\% \tag{15}$$

$$\omega_T = \frac{|\Delta R_T|}{|\Delta R_P| + |\Delta R_T| + |\Delta R_{climate\_interactive}| + |\Delta R_{human}|} \times 100\% \tag{16}$$

$$\omega_{climate\_interactive} = \frac{|\Delta R_{climate\_interactive}|}{|\Delta R_P| + |\Delta R_T| + |\Delta R_{climate\_interactive}| + |\Delta R_{human}|} \times 100\% \tag{17}$$

$$\omega_{human} = \frac{|\Delta R_{human}|}{|\Delta R_P| + |\Delta R_T| + |\Delta R_{climate\_interactive}| + |\Delta R_{human}|} \times 100\% \tag{18}$$



where $\omega_P$ (%), $\omega_T$ (%), $\omega_{climate\_interactive}$ (%), and $\omega_{human}$ (%), respectively,
represent the percentage contributions of precipitation change, temperature change,
interactions of climatic variables, and human activity to runoff change.

### 2.3.4 Hypothesized climate change scenarios

According to the projected change range of future climatic variables (Hoegh-Guldberg
et al., 2019; Zhang et al., 2022), we set four temperature change scenarios (T ± 0.5 ℃
and T ± 1 ℃) and four precipitation change scenarios (P ± 10% and P ± 20%) relative
to the temperature and precipitation during the period 1984–2018 to analyze the
responses to climate change of total runoff and runoff components. To study the
combined effect of simultaneous changes in precipitation and temperature on runoff,
the following four extreme combination scenarios were specially analyzed: scenario S1
(P − 20%, T − 1 ℃); scenario S2 (P − 20%, T + 1 ℃); scenario S3 (P + 20%, T − 1 ℃);
and scenario S4 (P + 20%, T + 1 ℃).

## 3 Results and discussion

### 3.1 Runoff simulation

Figure 2 shows the simulated and observed monthly runoff of the seven stations and
summarizes the model's performance. In general, the model achieved reasonably
satisfactory results, with *NSE* exceeding 0.68 at all stations. But there still existed a
certain degree of discrepancy between simulations and observations in some years for
some stations, such as when the simulation slightly overestimated runoff peaks at
Zhimenda Station in the validation period while it underestimated Changdu and
Xialaxiu Stations' lowest runoff in both the calibration and validation periods.
Uncertainty arising from model parameters and inferred runoff may have caused these
discrepancies, but the most likely and important reason may be the
over/underestimation of precipitation forcing (Miao et al., 2022; Su et al., 2016).

**Figure 2**

Accurate precipitation input is a prerequisite for obtaining reasonable model parameters





and simulation results (Zhang et al., 2013; Chen et al., 2017). Figure 3 compares the simulation results for monthly runoff driven by CMA precipitation and PERSIANN-CDR precipitation at Tuotuohe and Zhimenda Stations in the source area of the Yangtze River, where meteorological stations are extremely sparse (Figure 1). The CMA-precipitation-driven model exhibited poor performance even after parameter optimization, except for at Zhimenda Station during the calibration period. In contrast, taking *NSE* as an example, and in comparison with the result obtained using CMA precipitation, the *NSE* of Tuotuohe Station increased from 0.31 to 0.86 when PERSIANN-CDR precipitation was used during the validation period. Such a large improvement indicates that the PERSIANN-CDR precipitation, which describes the spatial variation of precipitation more accurately than the CMA precipitation obtained through very limited rain gauge interpolation in the source area of the Yangtze River (Liu et al., 2017b; Bai and Liu, 2018), can be used as input for the model to generate more accurate runoff. This sheds light on the importance of precipitation to hydrological research and the prospect of using PERSIANN-CDR precipitation products in alpine regions with sparse meteorological stations (Liu et al., 2017b). Because the model with CMA precipitation as input performed well in the source regions of the Yellow River and the Lancang River, there was no experiment using PERSIANN-CDR precipitation in these two headwater subregions.

**Figure 3**

### 3.2 Runoff components decomposition

Figure 4 shows the contributions of rainfall, snowmelt, and glacier runoff to the total annual runoff at the seven stations for 1984–2018. We found that rainfall runoff was the main component of runoff, accounting for 82%–92%, whereas snowmelt and glacier runoff both accounted for less than 10%, due to low temperatures and small glaciers (Table 1). Our current results are similar to the previous finding that snowmelt runoff and glacier runoff both make up a small proportion of the total runoff in the TRSR, but note that the specific values of their contributions to runoff differed among these studies. Wang et al. (2021b) found that during 1984–2015, snowmelt runoff contributed 15% to





the total runoff at Zhimenda Station. This estimate is higher than the 8.9% contribution
in 1984–2018 estimated by our current work and the 7% contribution in 2003–2014
estimated by Han et al. (2019). The discrepancies among these results may be attributed
to differences specific to the research periods, but various choices of forcing input data,
model parameters, and definitions of snowmelt runoff and glacier runoff should be more

important factors (Wang et al., 2021b; Zhao et al., 2019; Sun and Su, 2020).

**Figure 4**

Figure 5 shows the monthly variation in each runoff component. The regime of rainfall
runoff of all stations was highly consistent with that of total runoff, which once again
confirms the leading role of rainfall in total runoff. Snowmelt runoff mainly occurred

from April to June due to the melting of winter snowpack and spring snowfall, with
contributions ranging from 20% to 60% (Figure S2). As a result of higher altitude
(Figure 1) and lower temperature (Deng and Zhang, 2018), the snowmelt runoff peak
at Tuotuohe and Zhimenda Stations in the source area of the Yangtze River emerged in
June, while this peak occurred at the other stations in May. Glacier runoff was mainly

concentrated in July and August, corresponding to the higher temperatures during this
period (Figure S1). However, due to the fact that glaciers made up only a small
proportion of the total area (Table 1), the monthly contribution of glacier runoff was
below 25%.

**Figure 5**

**3.3 Runoff variation and attribution**

Compared with the reference period (1984–2003), the mean annual runoff in all sub-
basins of the TRSR during the changed period (2004–) increased (Figure 6a), especially
in the source area of the Yangtze River, with the catchments monitored by Zhimenda
and Tuotuohe Stations increasing by 31% and 51%, respectively. Figure 7 presents the

absolute impacts and relative contributions of four influencing factors (precipitation,
temperature, interactions of climatic variables, and human activity) on the annual runoff





increase in different sub-basins. It is clear that precipitation was strongly and positively correlated with the runoff, with relative contributions of 38%–71%, in agreement with the findings of Wu et al. (2018), who reported that increasing precipitation can directly
increase runoff. Because it has increased the most significantly in the source region of the Yangtze River during recent decades (Figure 6b), precipitation had a greater impact on runoff variation there than it did in the other two headwater subregions.

**Figure 6**

**Figure 7**

Although the runoff in catchments monitored by Tuotuohe and Xiangda Stations exhibited a minor increase (≤5%) when the temperature rose, warming temperature had an overall effect of reducing runoff (by 4%–25%), because the limited increase in meltwater was largely offset by the enhancement of evapotranspiration and the degradation of frozen soil (Zhao et al., 2019). Given the spatio-temporal changes in
climate variables and their complex interaction, the interactive effects of climate variables on runoff change were spatially non-uniform within the TRSR. Specifically, the absolute impacts of this factor on runoff change varied from −11 mm to 20 mm per year, and the relative contribution ranged from 0% to 26%. Human activity interference in the catchment monitored by Xialaxiu Station far exceeded that in other catchments.
It explains the 49% increase in runoff, which can be attributed to the significant degradation of grassland in this region during the changed period (Zeng et al., 2021; Zhang et al., 2021). In the context of increasing water demand related to agriculture and industry (Zhai et al., 2021) and the ecological protection policy proposed in the 21st century (Liu et al., 2017a), human activities have consistently reduced runoff in the
other catchments, except in the catchment monitored by Changdu Station.

Overall, the impacts of precipitation, temperature, the interactions of climatic variables, and human activities on runoff change present differences among different sub-basins.



Climate change, which integrates precipitation- and temperature-induced change with
that due to the interactions of climatic variables, accounted for over 75% of the change
in runoff for all basins except the catchment monitored by Xialaxiu Station, where the
contribution of climate change was 51%. Therefore, in the TRSR, the dominant factor
influencing runoff variation was climate change, especially precipitation.

**3.4 Hydrological responses to hypothetical climate change scenarios**

Figure 8 shows projected percent changes of total runoff and each runoff component
under four climate change combination scenarios with respect to the period from 1984
to 2018. The total runoff at all stations was expected to increase the most under scenario
S3 (P + 20%, T − 1 ℃), with an increase of 29%–80%, followed by scenario S4 (P +
20%, T + 1 ℃), with an increase of 25%–41%, whereas under scenarios S1 (P − 20%,
T − 1 ℃) and S2 (P − 20%, T + 1 ℃), respective decreases of 24%–34% and 21%–56%
were predicted. These results indicate that the total runoff was mainly affected by
precipitation, but the magnitude of response could strengthen or weaken with changing
temperature, which affects not only evapotranspiration but also meltwater (Su et al.,
2016). Spatially, as a consequence of neglected glacier runoff, total runoff changes at
Tangnaihai and Maqu Stations in the source region of the Yellow River were maximal
under scenario S3 in comparison with other stations. We further found that the total
runoff showed a higher sensitivity to precipitation increase than to precipitation
decrease when the temperature remained constant relative to the period of 1984–2018
(Figure S3). Taking Zhimenda Station as an example, the total runoff would increase
by 50% if precipitation increased by 20%, but decrease by about 41% if precipitation
decreased by 20%. This may be explained by the runoff generation process, just as
Spencer et al. (2019) have reported that a continuous multi-year pattern of lower- or
higher-than-average precipitation can reduce or fill basin storage and affect runoff
responses. The different behaviors of total runoff when only temperature changes are
shown in Figure S4. For the catchment monitored by Xiangda Station, total runoff was
basically unchanged when the temperature changed. This can be explained by the fact
that temperature has a similar degree of influence on evapotranspiration and meltwater





within this region, while evapotranspiration plays a stronger role in other sub-basins.
The pattern of change in rainfall runoff was essentially consistent with that of total
runoff, which is closely associated with precipitation although temperature change will
affect evapotranspiration.

**Figure 8**

As climatic variables changed, snowmelt runoff tended to have a larger degree of
variation (−76% to 203%) than did other runoff components, indicating that it is more
susceptible to climate change. Despite its similarity to rainfall runoff and total runoff,
snowmelt runoff also increased most obviously in scenario S3 (P + 20%, T − 1 ℃). The
major reasons for this were the direct increase in precipitation and the increased
proportion of precipitation falling as snow at the lower temperature (Chandel and Ghosh,
2021). Snowmelt runoff varied little in the wetter and warmer scenario S4 relative to
1984–2018, because increasing precipitation was basically entirely compensated for by
the decreased solid-to-liquid precipitation ratio. The most significant change in
snowmelt runoff was observed at Tuotuohe Station and can be explained by this
station's relatively high elevation, which amplifies the importance of snowmelt (Figure
1). Glacier runoff was highly dependent on temperature change whether precipitation
increased or decreased. Consistent with the small area proportion of glaciers in each
sub-basin (Table 1), the variation in glacier runoff did not show obvious spatial
heterogeneity.

Figure 9 presents the projected seasonal change percentage of total runoff under
different scenarios. As shown in Figures 9 and S5, on the seasonal scale, similar to the
annual scale, the change pattern of total runoff was similar to that of rainfall runoff,
further confirming the importance of rainfall runoff in the TRSR. Additionally, due to
over 80% of precipitation occurring in summer (Jul−Sep) and autumn (Oct−Dec)
(Figure S1), more obvious total runoff changes were projected during this period.
Although temperature was poorly and negatively correlated with the total runoff for
most stations at the seasonal scale, the total runoff at Tuotuohe Station increased by 10%

with a temperature increase of 1 °C in spring (Apr−Jun) as the result of the release of more meltwater, which could possibly advance the peak flow of snowmelt (Shen et al., 2018). Su et al. (2016) have also reported that in the source region of the Yangtze River,

an apparent earlier melt may happen in April under a warming climate. Due to the neglected glacier runoff and the more obvious warming trend compared with the other two source areas in the past few decades (Yi et al., 2011; Xie et al., 2004), the total runoff of the Yellow River source displayed a more strongly significant change (−18% to 25%) in summer when the temperature changed relative to 1984−2018 (Figure 9b).

Generally speaking, the snowmelt runoff has a second small peak because of the melting of fresh snowfall in autumn (Zhang et al., 2013), as well as the high temporal and spatial variability of precipitation in the source region of the Yangtze River (Liu et al., 2017b) leading to Tuotuohe Station's snowmelt runoff showing the highest sensitivity to climate change in autumn (Figure S6). The catchment monitored by

Xiangda Station has more glacier coverage (Table 1), which may be why its glacier runoff had a greater response to temperature in winter in comparison with the glacier runoff in the other basins (Figure S7).

**Figure 9**

In general, precipitation plays a greater role in total runoff and rainfall runoff, while

glacier runoff is dominated by temperature at annual and seasonal scales. Snowmelt runoff shows a different behavior. Annually, it is affected by both precipitation and temperature. In spring and summer, it is more related to precipitation, in winter it is more related to temperature, and in autumn it is more related to the combined effects of temperature and precipitation.

**3.5 Uncertainty**

This work provides a systematic understanding of hydrological processes in the TRSR. The results are encouraging, but some uncertainties are still worthy of further analysis in future research. We simulated glacier runoff using a simple degree-day factor algorithm, which is highly sensitive to its parameter DDF (Chen et al., 2017; Zhang et





al., 2013; Zhao et al., 2019). Given the lack of observed glacier information, we did not calibrate DDF, but directly adopted the constant values at basin scale from existing studies that did not take into account the high spatio-temporal heterogeneities of DDF (Zhang et al., 2006). Although the final simulation result was satisfactory, the practice of treating DDF as a uniform value still introduced a certain degree of bias to the results.

In fact, due to the very limited glacier melt water contribution (Figure 4), the uncertainty caused by glacier runoff is far less than that stemming from precipitation forcing data (Zhao et al., 2019). Many previous studies have reported that precipitation is the key factor limiting hydrological simulation performance in alpine regions with sparse meteorological stations (Liu et al., 2017b). Precipitation estimation error will bring

significant uncertainty to runoff simulations (Sun and Su, 2020). As shown in Figure 3, the model's performance was greatly improved by replacing CMA precipitation with PERSIANN-CDR data in the source region of the Yangtze River. This finding inspires us to hope that satellite-based precipitation products may be suitable alternatives for hydrological studies in alpine regions and that we can use several sets of precipitation

products to reduce the uncertainty of simulation as much as possible in the future.

   Some uncertainties may also exist in assessing the influence of climate change and human activities on changes in runoff. Hydrological model simulation splits the relationship and interaction between climate change and human activity, which

inevitably introduces a certain bias due to the complicated feedback and response relationship between them (Ji and Yuan, 2018a; Shi et al., 2022). And our study focused only on the effects of long-term mean annual precipitation and temperature change on runoff without considering the possible impact of the intra-annual fluctuations of these factors. Therefore, additional work should be carried out to examine the causes of runoff

change in more detail.

**4 Conclusions**

In this work, we evaluated the hydrological changes in the TRSR using the VIC-Glacier land surface hydrologic model. The main results are summarized as follows:



(1) The VIC-Glacier model achieved good performance in the TRSR, with $NSE$ above

0.68, but we must pay attention to the importance of accurate precipitation input for

successful simulation. In the source region of the Yangtze River, the $NSE$ of Zhimenda

Station was increased from 0.32 to 0.76 during the validation period by replacing CMA

precipitation with PERSIANN-CDR precipitation.

(2) The rainfall runoff played a dominant role in maintaining runoff for the TRSR

during 1984−2018, accounting for 82%−92% of the total, whereas snowmelt and

glacier runoff both contributed less than 10%. Seasonally, snowmelt runoff was mainly

concentrated between April and June, and the peak time for snowmelt runoff in the

source area of the Yangtze River was about one month later than in the other two

headwater sub-regions. Glacier runoff mainly occurred in July and August.

(3) Climate change was the main cause of runoff increase after 2004 in the TRSR, with

its percentage contribution reaching 75%−89%, except for in the catchment monitored

by Xialaxiu Station. More specifically, precipitation was the main climatic factor

leading to runoff change, especially in the source area of the Yangtze River.

(4) Through various hypothetical climate change scenario experiments, we found

snowmelt runoff was easily affected by the joint change of precipitation and

temperature at the annual scale and in autumn, while in spring and summer, it was more

subject to precipitation change and to temperature change in winter. Considering the

fact that a simultaneous increase of precipitation and temperature is the most likely

future climate change scenario, we expect that the total runoff, rainfall runoff, and

glacier runoff will increase in the future, while the snowmelt runoff will remain

basically unchanged. These insights could help decision makers allocate water

resources more rationally in the future.

**Acknowledgments**

This work was supported by the National Natural Science Foundation of China

(41877155, 42041006) and the State Key Laboratory of Earth Surface Processes and

Resource Ecology(2022-ZD-03). We are grateful for high-performance computing

support from the Center for Geodata and Analysis, Faculty of Geographical Science,



Beijing Normal University (https://gda.bnu.edu.cn/).

**Data availability**

Data set available on request to corresponding authors

**Author Contribution**

TS and CM designed and executed the hydrological modeling work. TS led the data analysis and wrote the initial draft of the paper. QD, JG, XG and XZ contributed

scientifically to the modeling and data analysis. All authors contributed to the paper by providing comments, editing, and suggestions.

**Competing Interests**

The authors declare no competing interests.

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



**Table 1. Characteristics of seven sub-basins in the Three-River Source Region.**

| Basin | Hydrological Station | Location | | Drainage area (km²) | Runoff data availability | Glacier area Proportion (%) |
|---|---|---|---|---|---|---|
| | | Latitude (°N) | Longitude (°E) | | | |
| Yellow | Tangnaihai | 35.50 | 100.15 | 121,972 | 1983–2018 | 0 |
| | Maqu | 33.96 | 102.08 | 86,048 | 1983–2018 | 0 |
| Yangtze | Tuotuohe | 34.22 | 92.44 | 15,924 | 1983–2017 | 1.1 |
| | Zhimenda | 33.02 | 97.23 | 137,704 | 1983–2018 | 0.82 |
| Lancang | Xialaxiu | 32.52 | 96.62 | 4,125 | 1983–2012 | 0.71 |
| | Xiangda | 32.13 | 96.61 | 17,909 | 1983–2016 | 1.3 |
| | Changdu | 31.15 | 97.18 | 54,228 | 1983–2010 | 0.45 |


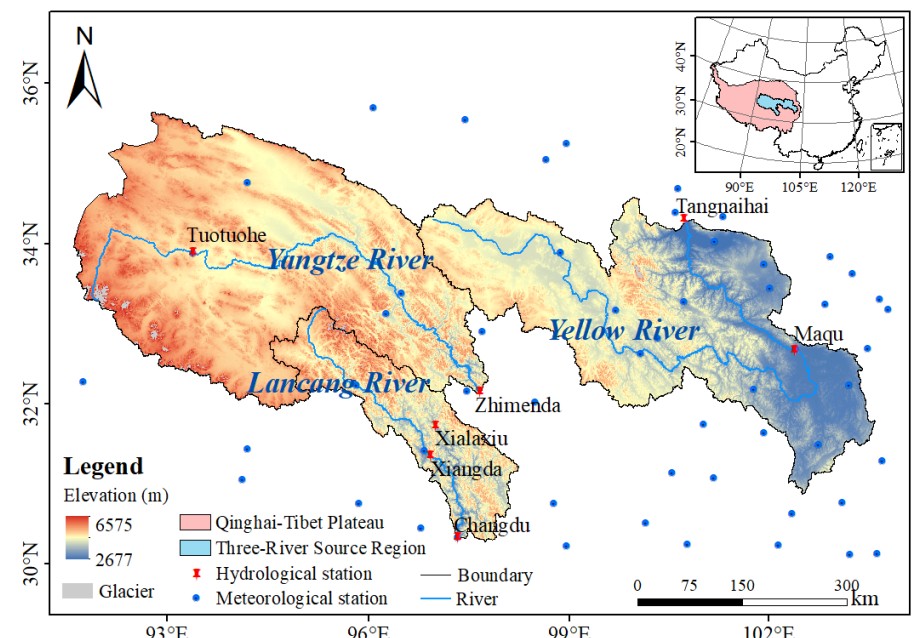

**Figure 1. Topography of the Three-River Source Region, distribution of glaciers, locations of meteorological and hydrological stations, basin boundaries, and river courses of the Yellow, Yangtze, and Lancang.**





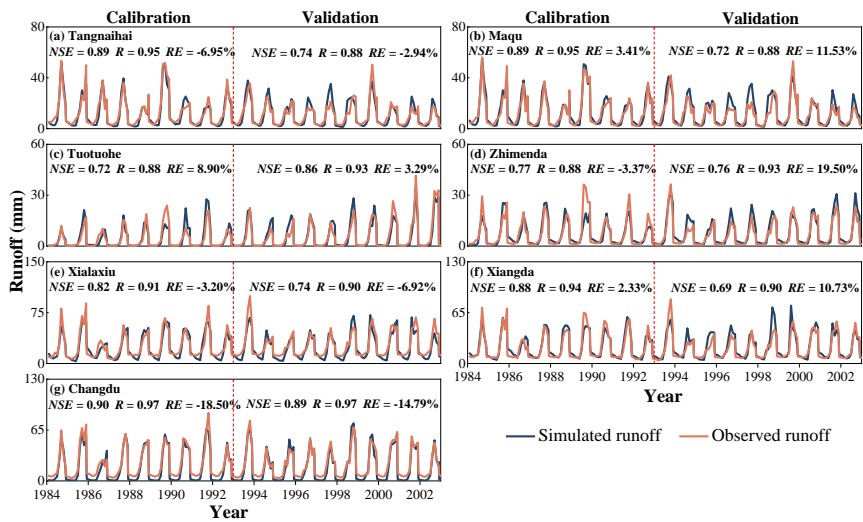

**Figure 2. Observed and simulated monthly runoff at the seven stations (a–g) during the calibration period (1984–1993) and validation period (1994–2003).**





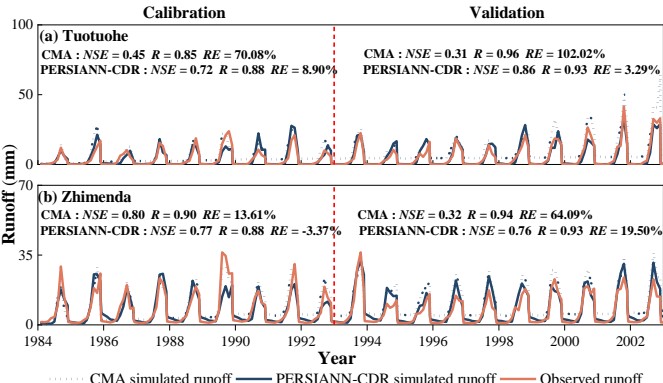

**Figure 3. Comparison of simulated monthly runoff based on CMA precipitation and PERSIANN-CDR precipitation during the calibration period (1984–1993) and validation period (1994–2003) for (a) Tuotuohe Station, and (b) Zhimenda Station.**






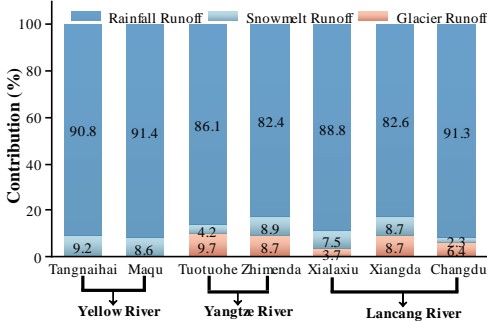

**Figure 4. Contributions of rainfall, snowmelt, and glacier runoff to the total annual runoff for the seven stations during 1984–2018. Numbers in the figure represent the relative contribution of each component.**






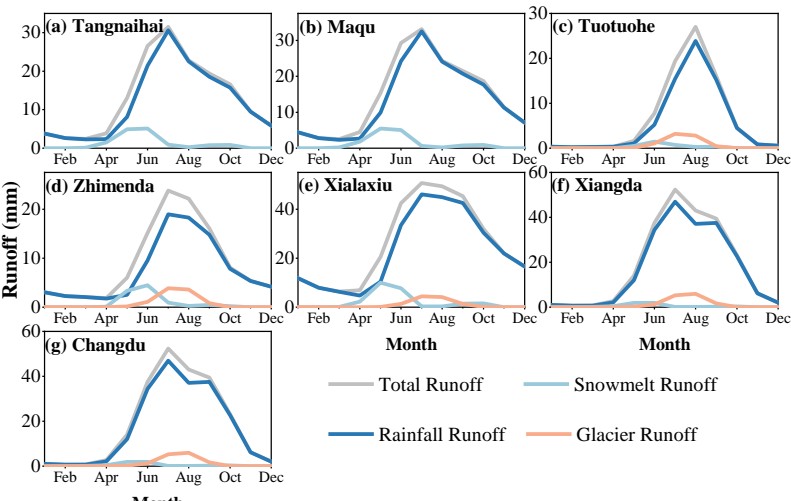

**Figure 5. Seasonal cycle of simulated total, rainfall, snowmelt and glacier runoff for the seven stations (a–g) during 1984–2018.**



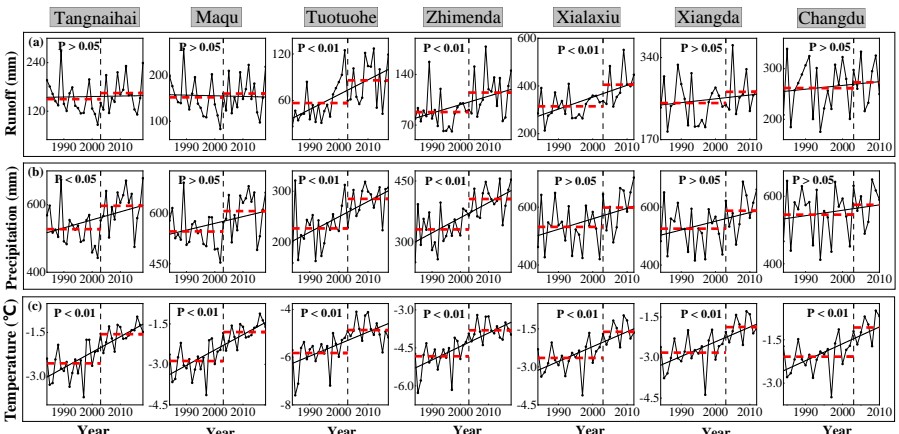

Figure 6. Annual runoff (a), precipitation(b), and temperature(c) for the seven sub-basins. The red dotted line represents the annual average of each variable in the reference period (1984–2003) and the changed period (2004–), respectively. *P* < 0.05 and *P* < 0.01 indicate that the trend is significant at the level of 0.05 and 0.01, respectively, using the Mann–Kendall trend test.





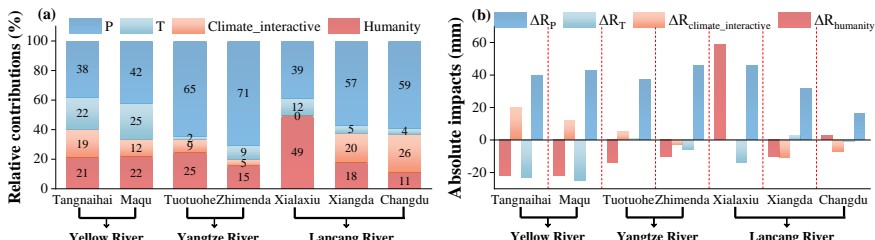

**Figure 7. Relative contributions (a) and absolute impacts (b) of the different influencing factors on the annual runoff trends at the seven stations. The numbers in (a) represent the relative contributions of the influencing factors as percentages.**






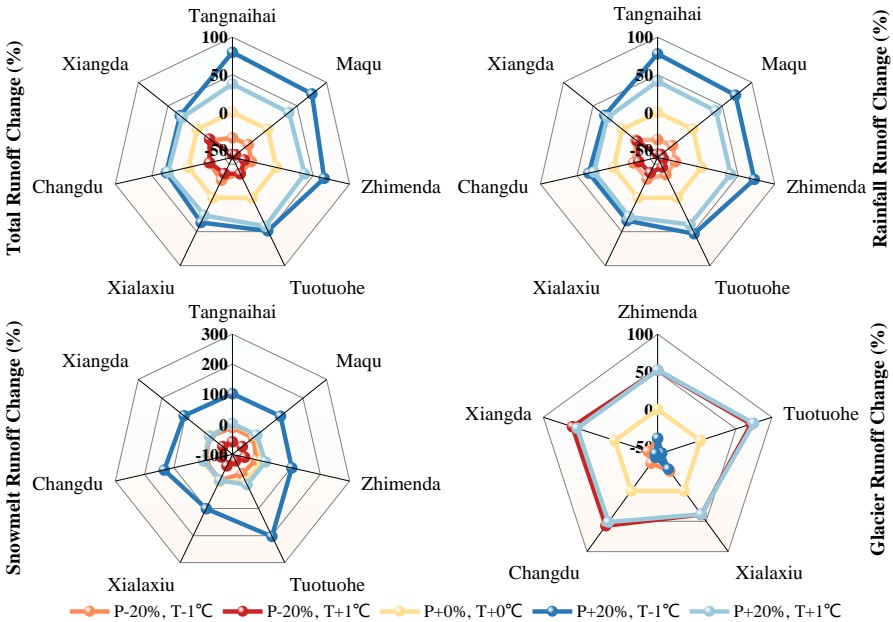

**Figure 8. Percentage changes in mean annual total, rainfall, snowmelt, and glacier runoff relative to the period 1984–2018 under four climate change combination scenarios for the seven stations.**






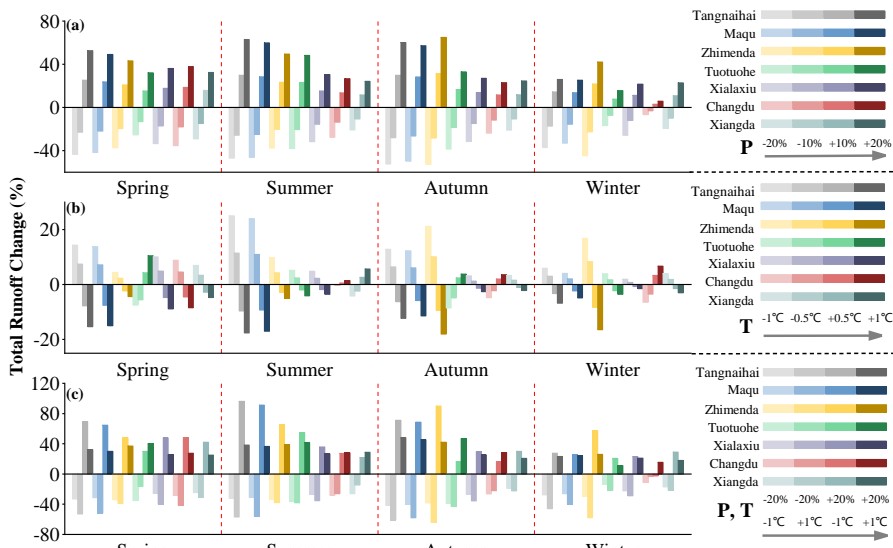

**Figure 9. Seasonal percentage change in total runoff relative to the period 1984–2018 under various scenarios for the seven stations. (a) Considering precipitation changes only; (b) Considering temperature changes only; (c) Considering precipitation and temperature changes simultaneously.**