# Peer review of "Hydrological response to climate change and human activities in the Three-River Source Region"

_Hydrology and Earth System Sciences, 2022_

## Author Comment (AC1)

**A DETAILED LIST OF RESPONSES**
**TO REVIEWER #1**

**General comments:**

It's my pleasure to review this manuscript. The TRSR region is important for water resource security and of interest to researchers because of its complex hydrological processes. This study conducts a systematic modeling work on this region, and analyze the contribution of runoff components and the hydrological response to climate change and human activities. The results are helpful for understanding the hydrological processes in this important region, which make this manuscript worth publishing. Overall, the manuscript is written well and easy to follow. However, I have some concern about the results, especially for the snow and glacier simulations. I recommend to accept the manuscript after moderate revisions to address following issues.

**Response:** We appreciate the positive assessment of our manuscript. Your insightful comments have enhanced our paper considerably. Below is a point-by-point response to your review.

**Specific comments:**

1. The description of model:
A module representing glacier processes was integrated into the model, and the authors described them in detail. The snowmelt contributes more than glacier runoff in most of the basins, but the simulation of snow processes was not introduced in the Method section. I think this might be due to that the snow module has been included in the VIC model, and the authors only introduced the extension module. Nonetheless, since the simulation of snow processes is equally important as glacier, I suggest the authors to add some description on the snow simulation.

**Response**: We thank the reviewer for this advice. We have added additional information about snow simulation to the revised manuscript, as follows:

"The critical elements of this model that are particularly relevant to its application in cold regions include (1) a two-layer energy-balance model that simulates accumulation and melt of ground snow and a simplified single-layer model of the ground snowpack energy balance that simulates melt, sublimation, drip and release of intercepted snow from the canopy (Cherkauer and Lettenmaier, 1999; Cherkauer and Lettenmaier, 2003; Storck and Lettenmaier, 1999); and (2) a frozen soil algorithm that calculates the soil ice contents within each vegetation type and the effects of frozen soil on infiltration and runoff (Cherkauer and Lettenmaier, 1999; Cherkauer and Lettenmaier, 2003)."

**References**
Cherkauer, K. A. and Lettenmaier, D. P.: Hydrologic effects of frozen soils in the upper

Mississippi River basin, J. Geophys. Res.-Atmos., 104, 19599-19610, https://doi.org/10.1029/1999jd900337, 1999.

Cherkauer, K. A. and Lettenmaier, D. P.: Simulation of spatial variability in snow and frozen soil, J. Geophys. Res.-Atmos., 108, https://doi.org/10.1029/2003jd003575, 2003.

Storck, P. and Lettenmaier, D. P.: Predicting the effect of a forest canopy on ground snow accumulation and ablation in maritime climates, 67th Western Snow Conference, edited by C. Troendle. Colo. State Univ., pp. 1–12.

2. Definition of the runoff component:

The authors estimated the contribution of runoff components in each basin, which is an important result. However, the result would be confusing if the definition of runoff component was not clarified. Is the runoff component defined based on the contribution of each water source in the total water input, or the proportion of each component in the streamflow? The amount of river water should be smaller than the sum of each water source due to evaporation loss. How does the model consider this? I suggest the authors to clearly clarify the definition of runoff components. The authors can refer to a recent review on this issue ("A meta-analysis based review of quantifying the contributions of runoff components to streamflow in glacierized basins").

**Response:** We thank the reviewer very much for this advice; it is indeed our negligence that the definition of runoff components is not clearly clarified, which would easily cause readers' misunderstanding. As He et al. (2021) stated, different definitions of runoff components could lead to different calculation results, and the same terminology in different studies might refer to different runoff components, thus preventing a comprehensive comparison of contributions of runoff components across different glacierized basins. In this study, the runoff component is defined as the proportion of each component in the streamflow, and the total runoff is divided into three components: glacier, rainfall, and snowmelt runoff. Glacier runoff represents the sum of glacier melt water and rainfall from the glacier area (Wang et al., 2021). Rainfall runoff represents the runoff induced by rainfall, and snowmelt runoff represents the runoff induced by snow melting. And we use the runoff simulated in the model when calculating contributions. Rainfall runoff and snowmelt runoff are calculated by the original VIC model, taking into consideration the contributions to infiltration and base flow as well as evapotranspiration losses, including canopy interception evaporation, vegetation transpiration, and bare soil evaporation. Glacier runoff is calculated by a degree-day algorithm, ignoring infiltration and evapotranspiration. We have edited the text of the method section of the revised manuscript to provide more detail, as follows:

"It is very important to define runoff components clearly (He et al., 2021). In this study, the runoff component is defined as the proportion of each component in the streamflow and the total runoff is divided into three components: glacier, rainfall, and snowmelt runoff. Glacier runoff represents the sum of glacier melt water and rainfall from glacier area (Wang et al., 2021). Rainfall runoff represents the runoff induced by rainfall and

snowmelt runoff represents the runoff induced by snow melting."

**References**

He, Z., Duethmann, D., and Tian, F.: A meta-analysis based review of quantifying the contributions of runoff components to streamflow in glacierized basins, J. Hydrol., 603, https://doi.org/10.1016/j.jhydrol.2021.126890, 2021.

Wang, Y., Xie, X., Shi, J., and Zhu, B.: Ensemble runoff modeling driven by multi-source precipitation products over the Tibetan Plateau, Chin. Sci. Bull., 66, 4169-4186, https://doi.org/10.1360/tb-2020-1557, 2021.

3. Validation of snow/glacier simulation:

It is good to involve snow and glacier simulation into the hydrological model, but the results could be unreasonable if the snow and glacier simulation are not validated by any measurement dataset. In my opinion, the contribution of glacier runoff in source Yangtze River (Zhimenda station) was significantly overestimated, and my approximate estimation is as follows: The mean annual runoff at Zhimenda station was about 160mm/a, so the glacier runoff should be 13.92mm/a (if the authors define the runoff component by the proportion in the streamflow). Considering the glacier area is 0.81%, the runoff generation in glacier area is 13.92/0.81%=1700mm/a. Excluding the precipitation (about 400mm/a), the glacier meltwater would be more than 1.3m/a, which is significantly higher than the estimation from existed glacier studies (0.5m/a). Besides, if the runoff component was defined by the water source definition, the glacier mass meltwater estimated in similar way would even be larger than 4m/a.

Nonetheless, I agree with the authors that the meltwater has little influence on the streamflow due to the small glacier area. But I just think that if snow and glacier simulations are not verified, the benefit of using a glacier hydrological model would be reduced.

**Response**: We thank the reviewer for these comments. We fully agree with the reviewer's assertion that the benefit of using a glacier hydrological model would be reduced if snow and glacier simulations are not verified. However, as to why snowmelt runoff and glacier runoff have not been verified, we make the following three responses:

(i) There is a lack of measured data of snowmelt and glacier runoff. Although some studies have calibrated and verified the snowmelt and glacier runoff, the data used in most studies are glacier outlines, snow cover area, or snow water equivalent derived from remote sensing data collected during several different time periods (Chen et al., 2017; Han et al., 2019; Sun and Su, 2020) rather than direct measured data of snowmelt and glacier runoff, which will inevitably bring some uncertainty to the simulation results (Zhao et al., 2019).

(ii) Actually, the observed total runoff includes glacier and snowmelt runoff; the simulation performance of snowmelt and glacier runoff can be reflected by evaluating the simulation results of VIC-Glacier on total runoff to a certain extent. As shown in Figure 2, the model achieved reasonably satisfactory results, with *NSE* exceeding 0.68

at all stations.

(iii) By comparing with previous studies, it is found that the research results on the proportion of glacier and snowmelt runoff in this study are within a reasonable range. Taking Zhimenda Station as an example, Wang et al. (2021) found that during 1984–2015, glacier runoff contributed 9% to the total runoff at Zhimenda Station; Han et al. (2019) and Zhang et al. (2013) estimated glacier runoff accounted for 5% and 6.5% in 2003–2014 and 1961–2009, respectively. These results are close to the conclusion in this study that glacier runoff accounts for 8.7% during 1984–2018 at Zhimenda Station.

Regarding the reviewer's concern that the contribution of glacier runoff as a source to the Yangtze River (Zhimenda Station) was significantly overestimated, we don't know which time period they are referring to during which the mean annual runoff at Zhimenda Station is about 160 mm/a, which is much higher than the 100 mm/a we obtained during 1984–2018. And the glacier runoff calculated based on the mean annual runoff of 100mm/a is 8.7 mm/a rather than 13.82 mm/a.

**References**

Chen, X., Long, D., Hong, Y., Zeng, C., and Yan, D.: Improved modeling of snow and glacier melting by a progressive two-stage calibration strategy with GRACE and multisource data: How snow and glacier meltwater contributes to the runoff of the Upper Brahmaputra River basin?, Water Resour. Res., 53, 2431-2466, https://doi.org/10.1002/2016wr019656, 2017.

Han, P., Long, D., Han, Z., Du, M., Dai, L., and Hao, X.: Improved understanding of snowmelt runoff from the headwaters of China's Yangtze River using remotely sensed snow products and hydrological modeling, Remote Sens. Environ., 224, 44-59, https://doi.org/10.1016/j.rse.2019.01.041, 2019.

Sun, H. and Su, F.: Precipitation correction and reconstruction for streamflow simulation based on 262 rain gauges in the upper Brahmaputra of southern Tibetan Plateau, J. Hydrol., 590, 125484, https://doi.org/10.1016/j.jhydrol.2020.125484, 2020.

Wang, Y., Xie, X., Shi, J., and Zhu, B.: Ensemble runoff modeling driven by multi-source precipitation products over the Tibetan Plateau, Chin. Sci. Bull., 66, 4169-4186, https://doi.org/10.1360/tb-2020-1557, 2021.

Zhang, L., Su, F., Yang, D., Hao, Z., and Tong, K.: Discharge regime and simulation for the upstream of major rivers over Tibetan Plateau, J. Geophys. Res.-Atmos., 118, 8500-8518, https://doi.org/10.1002/jgrd.50665, 2013.

Zhao, Q., Ding, Y., Wang, J., Gao, H., Zhang, S., Zhao, C., Xu, J., Han, H., and Shangguan, D.: Projecting climate change impacts on hydrological processes on the Tibetan Plateau with model calibration against the glacier inventory data and observed streamflow, J. Hydrol., 573, 60-81, https://doi.org/10.1016/j.jhydrol.2019.03.043, 2019.

4. Designation of climate change scenarios:
The authors set four scenarios to analyze the hydrological response to the climate

change. In my understanding, the scenarios designation seems more likely a sensitivity analysis between runoff and T and P, but the attribution analysis has shown the result that the precipitation is the most important factor. So we can expect the sensitivity analysis would give similar conclusion. If the aim of setting scenarios is to predict the runoff change in the future, why not directly use the projection climate data such as CMIP6?

**Response**: We very much appreciate these comments providing a different perspective related to the purpose of designating climate change scenarios. After discussing this point amongst the co-authors, we would prefer to retain original analysis for the following reasons:

(i) Hydrologic models driven by hypothetical climate change scenarios or global climate models (GCMs) have long been commonly used to predict the response of future runoff to climate change (Su et al., 2016).

(ii) The reliability of hydrological model simulation depends largely on the accuracy of meteorological forcing data (Sun and Su, 2020). Although GCMs have been greatly improved over time, their output cannot be directly applied to climate change prediction and related research at the watershed scale due to their inherent systematic deviation and coarse spatiotemporal resolution, so downscaling and deviation correction have become essential steps in using GCM output data, which would introduce uncertainty into the research results (Piani et al., 2010; Wood et al., 2004).

(iii) The hypothetical climate change scenarios are easy to design and apply; however, this does not mean that such scenarios can be assumed at will, but must be designed according to the possible range of future precipitation and temperature changes in previous studies. For example, Lutz et al. (2014) predicted the future precipitation and temperature of QTP would change $-10\%$ to $20\%$ and $1°C-3°C$ under the RCP4.5 and RCP8.5 scenarios from the CMIP5 multi-model ensemble, respectively. Su et al. (2016) estimated the annual precipitation would increase by $5.0-10.0\%$ in 2011–2040 and $10.0-20.0\%$ in 2041–2070 under RCP2.6, RCP4.5, and RCP8.5 scenarios at the plateau scale, and annual temperature was projected to increase for all scenarios, with the greatest warming in the northwest ($2.0-4.0$ °C) and least in the southeast ($1.2-2.8$ °C). Zhao et al. (2019) predicted the temperature of QTP would increase by 0.11 °C (for RCP2.6) and 0.31 °C (for RCP4.5) per decade. Extending these analyses, we chose precipitation changes from $-20\%$ to $+20\%$ at a step of $10\%$ and temperature changes from $-1°C$ to $1°C$ at a step of $0.5$ °C to analyze the impact of future climate change on runoff.

(iv) Sensitivity analysis is often used to identify the governing factors for a certain process simulated (Xu et al., 2004). Actually, without sensitivity analysis between climate factors (e.g., precipitation and temperature) and runoff, we can still roughly know the relationship between them, such as the fact that rainfall runoff is subject to precipitation change and glacier runoff depends on temperature change. Hypothetical climate change scenarios are made for the purpose of predicting how runoff will change in the future according to the possible range of future climate factors, which is exactly their practical significance. As mentioned in the conclusion of the original manuscript,

"Considering the fact that a simultaneous increase of precipitation and temperature is the most likely future climate change scenario, we expect that the total runoff, rainfall runoff, and glacier runoff will increase in the future, while the snowmelt runoff will remain basically unchanged." If this part is taken as sensitivity analysis, we could analyze only how runoff will change with precipitation and temperature change, instead of discussing that the most likely climate change scenario in the future is warming and wetting, and we would not draw a conclusion about how runoff will change in the future. Therefore, we think these hypothetical climate change scenarios are not just simple sensitivity analysis, but can be used to predict future runoff changes to a certain extent.

**References**

Lutz, A. F., Immerzeel, W. W., Shrestha, A. B., and Bierkens, M. F. P.: Consistent increase in High Asia's runoff due to increasing glacier melt and precipitation, Nat. Clim. Change, 4, 587-592, https://doi.org/10.1038/nclimate2237, 2014.

Piani, C., Weedon, G. P., Best, M., Gomes, S. M., Viterbo, P., Hagemann, S., and Haerter, J. O.: Statistical bias correction of global simulated daily precipitation and temperature for the application of hydrological models, J. Hydrol., 395, 199-215, https://doi.org/10.1016/j.jhydrol.2010.10.024, 2010.

Su, F., Zhang, L., Ou, T., Chen, D., Yao, T., Tong, K., and Qi, Y.: Hydrological response to future climate changes for the major upstream river basins in the Tibetan Plateau, Global Planet. Change, 136, 82-95, https://doi.org/10.1016/j.gloplacha.2015.10.012, 2016.

Sun, H. and Su, F.: Precipitation correction and reconstruction for streamflow simulation based on 262 rain gauges in the upper Brahmaputra of southern Tibetan Plateau, J. Hydrol., 590, 125484, https://doi.org/10.1016/j.jhydrol.2020.125484, 2020.

Wood, A. W., Leung, L. R., Sridhar, V., and Lettenmaier, D. P.: Hydrologic implications of dynamical and statistical approaches to downscaling climate model outputs, Clim. Change, 62, 189-216, https://doi.org/10.1023/B:CLIM.0000013685.99609.9e, 2004.

Xu, C., Hu, Y., Chang, Y., Jiang, Y., Li, X., Bu, R., and He, H.: Sensitivity analysis in ecological modeling, J. Appl. Ecol., 15, 1056-1062, 2004.

Zhao, Q., Ding, Y., Wang, J., Gao, H., Zhang, S., Zhao, C., Xu, J., Han, H., and Shangguan, D.: Projecting climate change impacts on hydrological processes on the Tibetan Plateau with model calibration against the glacier inventory data and observed streamflow, J. Hydrol., 573, 60-81, https://doi.org/10.1016/j.jhydrol.2019.03.043, 2019.

---

## Author Comment (AC2)

**A DETAILED LIST OF RESPONSES**
**TO REVIEWER #2**

**General comments:**

This study tried to quantify the contributions from different runoff component (rainfall, snowmelt and glacier runoff) to the total runoff based on a well-established VIC-Glacier model, and then discussed the potential causes resulting in the runoff changes. Besides the traditional ground-based hydro-meteorological observation, the remote sensed precipitation was also involved. In general, the paper is well-structured with the methods and results clearly presented, and the findings are attractive. I would like to suggest a few aspects for improving and making the statements and results more robust. Below please find my detailed comments on this work:

**Response:** We would like to thank you for your valuable comments on our paper. Your insightful review has enhanced our paper considerably. Below is a point-by-point response to your comments.

**Specific comments:**

"···affects the water resources security of 700 million people…", 700 million? Please double the number.

**Response:** We thank the reviewer for this advice. Once again, we carefully consulted the relevant literature, which indicated that the Three-River Source Region affects the water resources security of 700 million people living downstream (Ji et al., 2020).

**References**
Ji, P., Yuan, X., Ma, F., and Pan, M.: Accelerated hydrological cycle over the Sanjiangyuan region induces more streamflow extremes at different global warming levels, Hydrol. Earth Syst. Sci., 24, 5439-5451, https://doi.org/10.5194/hess-24-5439-2020, 2020.

Actually some previous studies have employed the VIC-Glacier to simulate the hydrological process in high mountain. The author need to review and summarize them in the introduction.

**Response:** We thank the reviewer for this advice. We have added some details about using VIC-Glacier to simulate QTP's hydrological processes to the revised manuscript, as follows:

"Hydrological models have been widely used to study the runoff response to climate changes in basins of QTP (Lutz et al., 2014; Su et al., 2016; Zhang et al.,2013). Wang

et al. (2021), using the variable infiltration capacity (VIC) land surface hydrologic model linked with the degree-day factor algorithm (VIC-Glacier), reported the total runoff of QTP showed an increasing trend during 1984–2015, and the glacier runoff in the south increased rapidly at a rate of 6 mm/a. Zhang et al. (2013) applied the VIC-Glacier model to quantify the proportion of rainfall, snowmelt and glacier runoff in six major basins of QTP during 1961–2009. On the basis of the work of Zhang et al. (2013), Su et al. (2016) applied the same VIC-Glacier model to determine the total runoff of QTP's six basins in 2041–2070 would increase by 2.7–22.4% relative to 1971–2000 due to increased rainfall runoff in the upstream of the Yellow, Yangtze, Salween, and Mekong Rivers and increased glacier meltwater of the upper Indus. However, some studies have shown that the rapid retreat of glaciers caused by climate warming would eventually reduce the water supply in the glaciated regions (Zhao et al., 2019; Barnett et al., 2005)."

**References**

Barnett, T. P., Adam, J. C., and Lettenmaier, D. P.: Potential impacts of a warming climate on water availability in snow-dominated regions, Nature, 438, 303-309, https://doi.org/10.1038/nature04141, 2005.

Lutz, A. F., Immerzeel, W. W., Shrestha, A. B., and Bierkens, M. F. P.: Consistent increase in High Asia's runoff due to increasing glacier melt and precipitation, Nat. Clim. Change, 4, 587-592, https://doi.org/10.1038/nclimate2237, 2014.

Su, F., Zhang, L., Ou, T., Chen, D., Yao, T., Tong, K., and Qi, Y.: Hydrological response to future climate changes for the major upstream river basins in the Tibetan Plateau, Global Planet. Change, 136, 82-95, https://doi.org/10.1016/j.gloplacha.2015.10.012, 2016.

Wang, Y., Xie, X., Shi, J., and Zhu, B.: Ensemble runoff modeling driven by multi-source precipitation products over the Tibetan Plateau, Chin. Sci. Bull., 66, 4169-4186, https://doi.org/10.1360/tb-2020-1557, 2021.

Zhang, L., Su, F., Yang, D., Hao, Z., and Tong, K.: Discharge regime and simulation for the upstream of major rivers over Tibetan Plateau, J. Geophys. Res.-Atmos., 118, 8500-8518, https://doi.org/10.1002/jgrd.50665, 2013.

Zhao, Q., Ding, Y., Wang, J., Gao, H., Zhang, S., Zhao, C., Xu, J., Han, H., and Shangguan, D.: Projecting climate change impacts on hydrological processes on the Tibetan Plateau with model calibration against the glacier inventory data and observed streamflow, J. Hydrol., 573, 60-81, https://doi.org/10.1016/j.jhydrol.2019.03.043, 2019.

Data, besides the PERSIANN-CDR, maybe the author could try proxy precipitation data as well.

**Response:** Thank you for this comment. Although the use of different precipitation sources, including proxy precipitation data, is an important topic in the hydrological research of the Three-River Source Region, we believe a deeper discussion of this point is beyond the scope of this manuscript. This is because the main purpose of this study

is to study runoff, not to compare the effects of different precipitation input data on runoff simulation. The precipitation data can be considered to be suitable for a specific study area, as long as the runoff simulation results meet the relevant evaluation criteria (e.g., *NSE*, *RE,* and *R*) by optimizing parameters. And previous studies have also confirmed that PERSIANN-CDR is a high-quality precipitation data set applicable to the source region of the Yangtze River (Liu et al., 2017; Wang et al., 2021). Therefore, in this study, when the VIC-Glacier model could not achieve reasonably satisfactory results using CMA precipitation in the source region of the Yangtze River, we used PERSIANN-CDR precipitation data. Perhaps we could carry out related research on the influence of different precipitation data sets on runoff simulation in a future study.

**References**

Liu, X., Yang, T., Hsu, K., Liu, C., and Sorooshian, S.: Evaluating the streamflow simulation capability of PERSIANN-CDR daily rainfall products in two river basins on the Tibetan Plateau, Hydrol. Earth Syst. Sci., 21, 169-181, https://doi.org/10.5194/hess-21-169-2017, 2017

Wang, Y., Xie, X., Shi, J., and Zhu, B.: Ensemble runoff modeling driven by multi-source precipitation products over the Tibetan Plateau, Chin. Sci. Bull., 66, 4169-4186, https://doi.org/10.1360/tb-2020-1557, 2021.

The glacier area plays a critical role in the whole hydrological simulation, the involved glacier area data in this study is multi-year average or just observed in 2017? I suggest to provide more detailed information.

**Response**: We are grateful to the reviewer for this suggestion. We see that this part was not explained in detail in the original submission, and we thank the reviewer for pointing this out. Firstly, we answer the reviewer's question, the glacier area data is a multi-year average for the mid-1970s. To make this clearer, we have added to the description of the glacier area data as follows:

"Glacier area data were from TPG1976 generated by Ye et al. (2017). This data set was specially compiled for QTP based on Landsat satellite images from the mid-1970s, among which most images were acquired from 1976, and SRTM digital elevation models (DEM v4.1) and Google Earth imagery."

**References**

Ye, Q., Zong, J., Tian, L., Cogley, J. G., Song, C., and Guo, W.: Glacier changes on the Tibetan Plateau derived from Landsat imagery: mid-1970s-2000-13, J. Glaciol., 63, 273-287, https://doi.org/10.1017/jog.2016.137, 2017.

For the missing observed runoff from Nov. to Apr. at Tuotuohe, it is suggested to discuss the influence (or uncertainty) in the runoff simulation and the corresponding contribution.

**Response:** We sincerely thank the reviewer for pointing out this issue. Although it was reasonable to treat the runoff of Tuotuohe Station from November to April as 0 because the runoff was negligible (Ahmed et al., 2020; Luo et al., 2019), it still introduced some uncertainty. The lack of data affected the model evaluation metrics, thus affecting the optimization results of model parameters. Thus we have added discussion about this uncertainty in section 3.5 of our revised manuscript, as follows:

"Due to the lack of observed runoff in Tuotuohe Station from November to April, we treated the runoff as 0. Although previous studies have shown that the runoff in these months is negligible (Ahmed et al., 2020; Luo et al., 2019), it inevitably introduced some uncertainty, because the missing data would affect the model evaluation metrics, thus affecting the optimization results of model parameters."

**References**

Ahmed, N., Wang, G., Booij, M. J., Oluwafemi, A., Hashmi, M. Z.-u.-R., Ali, S., and Munir, S.: Climatic Variability and Periodicity for Upstream Sub-Basins of the Yangtze River, China, Water, 12, 842, https://doi.org/10.3390/w12030842, 2020.

Luo, Y., Qin, N., Zhou, B., Li, J., Liu, J., Wang, C., and Pang, Y.: Change of Runoff in the Source Regions of the Yangtze River from 1961 to 2016, Res. Soil Water Conserv., 26, 123-128, 2019.

Attribution analysis, why the author subjectively divides the runoff time series into pre- and post-2003 periods? Why 2003 but not 2000 or 2005?

**Response**: We thank the reviewer for raising this issue; it is indeed our negligence that the runoff time series division of attribution analysis is not clarified in the original manuscript, which could be a gap for the reader. Therefore, we made the following modifications in the method part of the revised manuscript:

"Ecological restoration is the dominant anthropogenic interference with TRSR in recent decades (Feng et al., 2017). Chinese government classified TRSR as a national nature reserve in 2003, and in 2005, the government invested 7.5 billion yuan (RMB) to carry out an ecological protection project (Ma et al., 2021; Zhai et al., 2021). These measures suggest that disturbance by human activities in the TRSR increased notably around 2003. In view of this, we regard 1984–2003 as our reference period, with less human activity, while the period after 2004 is taken as the changed period (2004–), with a greater effect of human activities on runoff."

**References**

Feng, A., Li, Y., Gao, J., Wu, S., and Feng, A.: The determinants of streamflow variability and variation in Three-River Source of China: climate change or ecological restoration?, Environ. Earth Sci., 76, https://doi.org/10.1007/s12665-017-7026-6, 2017.

Ma, L., Liu, Z., Zhao, B., Lyu, J., Zheng, F., Xu, W., and Gan, X.: Variations of runoff

and sediment and their response to human activities in the source region of the Yellow River, China, Environ. Earth Sci., 80, https://doi.org/10.1007/s12665-021-09850-w, 2021.

Zhai, X., Yan, C., Xing, X., Jia, H., Wei, X., and Feng, K.: Spatial-temporal changes and driving forces of aeolian desertification of grassland in the Sanjiangyuan region from 1975 to 2015 based on the analysis of Landsat images, Environ. Monit. Assess., 193, https://doi.org/10.1007/s10661-020-08763-8, 2021.

The residual error in the equations 10-12 could also come from the observational error (including climatic forcing data or measured runoff)

$$\varepsilon = Rr_{sim} - Rr_{obs}$$

**Response**: We fully agree with this comment and provide clarification as follows:

"$\varepsilon$ is the residual error, which may be related to factors not considered in this study, such as model simulation error and observational error, including climatic forcing data or observed runoff."

Lines 285, does the "interactions of climatic variables" mean the interaction between precipitating and temperature?

**Response**: We thank the reviewer for raising this issue. Due to the complex interactions of climatic factors in hydrological processes, the sum of the influences of each single climate factor change on runoff was not equal to the runoff change caused by all the climate factors changing at the same time. Another relevant factor is that the influence of wind speed on runoff change was not discriminated. So in this study, we recorded these unconsidered factors as the interactions of climatic variables. Therefore, the interactions of climatic variables included not only the interaction between precipitation and temperature, but also the interactions between precipitation and wind speed and between temperature and wind speed, as well as between precipitation, temperature, and wind speed.

Hypothesized climate change scenarios, although the author use "hypothesized" to explain the virtual state, it is preferred to illuminate them as sensitivity analysis in my opinion.

**Response:** We very much appreciate these comments providing a different perspective related to the purpose of climate change scenario design. After discussing this point amongst the co-authors, we would prefer to retain original analysis for the following reasons:
(i) Using hypothetical climate change scenarios to drive hydrological models is one of the methods that can be used to analyze hydrological responses to climate change (Su et al., 2016).
(ii) Hypothetical climate change scenarios are set according to the possible range of

future precipitation and temperature changes in previous studies. Lutz et al. (2014) predicted the future precipitation and temperature of QTP would change −10% to 20% and 1°C–3°C under the RCP4.5 and RCP8.5 scenarios from the CMIP5 multi-model ensemble, respectively. Su et al. (2016) estimated the annual precipitation would increase by 5.0–10.0% in 2011–2040 and 10.0–20.0% in 2041–2070 under the RCP2.6, RCP4.5, and RCP8.5 scenarios at the plateau scale, and annual temperature is projected to increase for all scenarios, with the greatest warming in the northwest (2.0–4.0 °C) and least in the southeast (1.2–2.8 °C). Zhao et al. (2019) predicted the temperature of QTP would increase by 0.11 °C (for RCP2.6) and 0.31 °C (for RCP4.5) per decade. Extending these analyses, we chose precipitation changes from −20% to +20% at a step of 10% and temperature changes from −1°C to 1°C at a step of 0.5 °C to analyze the impact of future climate change on runoff.

(iii) Sensitivity analysis is often used to identify the governing factors for a certain process simulated (Xu et al., 2004). Actually, without sensitivity analysis between climate factors (e.g., precipitation, and temperature) and runoff, we can still roughly know the relationship between them, such as the fact that rainfall runoff is subject to precipitation change and glacier runoff depends on temperature change. Hypothetical climate change scenarios are made for the purpose of predicting how runoff will change in the future according to the possible range of future climate factors, which is exactly their practical significance. As mentioned in the conclusion of the original manuscript, "Considering the fact that a simultaneous increase of precipitation and temperature is the most likely future climate change scenario, we expect that the total runoff, rainfall runoff, and glacier runoff will increase in the future, while the snowmelt runoff will remain basically unchanged." If this part is taken as sensitivity analysis, we could analyze only how runoff will change with precipitation and temperature change, instead of discussing that the most likely climate change scenario in the future is warming and wetting, and we would not draw a conclusion about how runoff will change in the future. Therefore, we think these hypothetical climate change scenarios are not just simple sensitivity analysis, but can be used to predict future runoff changes to a certain extent.

**References**

Lutz, A. F., Immerzeel, W. W., Shrestha, A. B., and Bierkens, M. F. P.: Consistent increase in High Asia's runoff due to increasing glacier melt and precipitation, Nat. Clim. Change, 4, 587-592, https://doi.org/10.1038/nclimate2237, 2014.

Su, F., Zhang, L., Ou, T., Chen, D., Yao, T., Tong, K., and Qi, Y.: Hydrological response to future climate changes for the major upstream river basins in the Tibetan Plateau, Global Planet. Change, 136, 82-95, https://doi.org/10.1016/j.gloplacha.2015.10.012, 2016.

Xu, C., Hu, Y., Chang, Y., Jiang, Y., Li, X., Bu, R., and He, H.: Sensitivity analysis in ecological modeling, J. Appl. Ecol., 15, 1056-1062, 2004.

Zhao, Q., Ding, Y., Wang, J., Gao, H., Zhang, S., Zhao, C., Xu, J., Han, H., and Shangguan, D.: Projecting climate change impacts on hydrological processes on the Tibetan Plateau with model calibration against the glacier inventory data and observed streamflow, J. Hydrol., 573, 60-81,

https://doi.org/10.1016/j.jhydrol.2019.03.043, 2019.

Figure 3, did the simulation here use the same optimized parameters?

**Response**: In this study, we separately calibrated and optimized VIC-Glacier model in the source region of Yangtze River using CMA precipitation and PERSIANN-CDR precipitation, respectively. So there were two sets of different optimized parameters. Previous literature has indicated that hydrologic model recalibration is important when applying different precipitation products (Camici et al., 2018; Ciabatta et al., 2016; Khan et al., 2011).

**References**

Camici, S., Ciabatta, L., Massari, C., and Brocca, L.: How reliable are satellite precipitation estimates for driving hydrological models: A verification study over the Mediterranean area, J. Hydrol., 563, 950-961, https://doi.org/10.1016/j.jhydrol.2018.06.067, 2018.

Ciabatta, L., Brocca, L., Massari, C., Moramarco, T., Gabellani, S., Puca, S., and Wagner, W.: Rainfall-runoff modelling by using SM2RAIN-derived and state-of-the-art satellite rainfall products over Italy, Int. J. Appl. Earth Obs. Geoinf., 48, 163-173, https://doi.org/10.1016/j.jag.2015.10.004, 2016.

Khan, S. I., Hong, Y., Wang, J., Yilmaz, K. K., Gourley, J. J., Adler, R. F., Brakenridge, G. R., Policelli, F., Habib, S., and Irwin, D.: Satellite Remote Sensing and Hydrologic Modeling for Flood Inundation Mapping in Lake Victoria Basin: Implications for Hydrologic Prediction in Ungauged Basins, IEEE Trans. Geosci. Remote Sens., 49, 85-95, https://doi.org/10.1109/tgrs.2010.2057513, 2011.

Table 1, I noticed that the lengths of available runoff data varies among the station, how to deal with the different time series during the calculation process. In addition, some results should not correspond to 1984–2018, e.g. in the Figures 4 and 5.

**Response**: We thank the reviewer for these comments. We indeed collected available runoff data of different lengths among the stations, but this did not affect this study. Actually, only a certain period of measured runoff data that was less affected by human beings was needed to evaluate the performance of the model when using the VIC-Glacier model to simulate runoff, and then we could apply the optimized model to simulate runoff at any time, as long as there is model input data for the corresponding period. In this study, we set 1983 as the warm-up period for the model, 1984–1993 as the calibration period, and 1994–2003 as the validation period. So we needed the measured runoff data only from 1983 to 2003. It was based on simulated runoff when calculating the proportion of runoff components and hydrological response to hypothetical climate change scenarios, so we chose 1984–2018 as the research period. For attribution analysis, the measured runoff data of the whole research period were needed, but since different stations have different lengths of runoff data, we divided the research period into pre- and post-2003. For example, the research period was divided

into 1984–2003 and 2004–2010 for Changdu Station, while for Tangnaihai Station, it was divided into 1984–2003 and 2004–2018.

Figure 4, how to tell the liquid (corresponding rainfall runoff) and solid precipitation (corresponding snowmelt runoff) from the observed data?

**Response**: Thanks for this question. Precipitation can be automatically partitioned into solid and liquid components when precipitation events occur according to the temperature threshold set when using VIC model. In this study, we set the maximum air temperature at which snowfall occurs and minimum air temperature at which rainfall occurs to be 1.5°C and –0.5°C, respectively, based on expert‐recommended temperature thresholds from the VIC website (https://vic.readthedocs.io/en/master/).

Figure 5, the total runoff represent the total surface runoff? Does it include the baseflow?

**Response**: We agree that the relevant information on this issue is not clearly stated in the original submission. The total runoff is the sum of surface runoff and baseflow. We have supplemented the text with statements about surface runoff and base flow in the method section, as follows:

"The VIC model divides the soil column of each grid cell into three layers. The surface runoff generated from the upper two soil layers is simulated based on the variable soil moisture capacity curve, and the base flow generated from the third layer based on the nonlinear ARNO model (Todini, 1996)."

**References**
Todini, E.: The ARNO rainfall-runoff model, J. Hydrol., 175, 339-382, https://doi.org/10.1016/s0022-1694(96)80016-3, 1996.

Figure S1, but the author mentioned them as 1983–2018 in Lines 146?

**Response**: Thank the reviewer for this comment. In this study, the meteorological data was obtained from 1983 to 2018, but we set 1983 as the warm-up period to get an optimal initial state of VIC-Glacier model simulation, so we did not analyze the research results and meteorological data of 1983.